# GLIM: Towards Generalizable Learning Representation for MILP

## Abstract

Mixed-Integer Linear Programs (MILPs) underpin a wide range of combinatorial optimization applications, and there has been many works in the field of MILPs based on machine learning. However, existing learning-based approaches often struggle to generalize beyond narrow training distributions or specific tasks. In this paper, we introduce **GLIM** (Generalizable Learning Representation for MILPs), a general-purpose embedding model designed to unify learning across diverse MILP classes and downstream tasks. GLIM is trained on a large corpus of roughly 78,000 instances spanning 2,000 problem classes. Motivated by the observation that *problem type* and *problem scale* are orthogonal factors whose interaction drives empirical difficulty, GLIM learns a joint representation that disentangles type, scale, and solving complexity. Each instance is encoded as a bipartite graph and processed by a hybrid architecture that couples GNN modules with Perceiver-like blocks. We evaluate GLIM on two representative MILP tasks to probe representation quality: (i) *MILP Instance Retrieval* and (ii) *MILP Solver Hyperparameter Prediction*. Across in-distribution and distribution-shifted settings, including real-world MIPLIB benchmarks, GLIM outperforms strong baselines in most cases and exhibits robust transfer to new classes and sizes. These results indicate that a single, disentangled embedding can serve as a reusable backbone for MILP tasks, enabling broader generalization than task- or class-specific learned components.

## 1 Introduction

Mixed-Integer Linear Programs (MILPs) are a central modeling tool for combinatorial optimization, with applications spanning logistics (Song et al., 2018), scheduling (Bradac et al., 2014), planning (Ren & Gao, 2010), and supply chain management (Soylu et al., 2006). Despite decades of progress in exact and heuristic algorithms, large-scale MILPs remain computationally challenging due to their NP-hardness. Beyond exact solvers such as Gurobi (Gurobi Optimization, LLC, 2024), CPLEX (Holmström et al., 2009), and SCIP (Bolusani et al., 2024), recent studies have explored machine learning (ML) techniques to accelerate MILP solving. Learning-based methods have been proposed for branching (Gupta et al., 2022; 2020), cutting-plane selection (Tang et al., 2020; Wang et al., 2023), and large neighborhood search (Wu et al., 2021; Ye et al., 2025), showing that ML can exploit structural patterns in MILPs and improve over purely hand-crafted heuristics. However, these approaches often face limited generalization. On the *data level*, models trained on one distribution (e.g., Set Cover) typically transfer poorly to related variants or to other classes such as Facility Location. On the *task level*, many learned components are tied to specific solver subroutines, making them brittle under distribution shift and hard to reuse across tasks.

These limitations highlight a broader challenge: while task-specific learning can yield improvements, it does not scale to the diversity of MILP problems encountered in practice. A natural next step is to seek *unified models* that can capture common structure across heterogeneous problem classes and provide reusable representations for multiple tasks. Recent efforts in this direction (Zong et al., 2025; Drakulic et al., 2024; Pan et al., 2025; Cai et al., 2025) are promising, as they aim to train a single model across multiple problem classes for solving tasks. However, these approaches are typically trained and evaluated only on a few predefined problem types (e.g., graph decision problems or variants of the Traveling Salesman Problem), making it difficult to extend them to other problem types. Consequently, they cannot directly generalize to broader settings, which limits their

applicability in real-world scenarios where both instance distributions and task requirements are highly diverse.

This motivates a central research question: *Can we train a unified model that generalizes across diverse MILP distributions and supports multiple downstream tasks?* Embedding models offer a practical path toward such unification. In other modalities such as natural language (Devlin et al., 2019; Zhang et al., 2025), vision (Dosovitskiy et al., 2020; Radford et al., 2021), and audio (Baevski et al., 2020), embedding models achieve broad generalization by mapping heterogeneous inputs into a shared representation space that is reusable across tasks. By analogy, an appropriately designed MILP embedding model could serve as a reusable backbone for diverse data regimes and downstream objectives.

In this work, we introduce **GLIM** (Generalizable Learning Representation for MILPs), an embedding model tailored to the multifaceted nature of MILP instances. Compared to existing work, GLIM incorporates three novel yet complementary strategies. (i) We construct a large-scale, highly diverse training corpus of approximately 70,000 instances spanning 2,000 problem classes, extended from existing MILP generation pipelines. (ii) We encode each MILP as a bipartite graph and process it with a hybrid architecture that integrates GNN modules with Perceiver-like blocks, adapted to accommodate large input sizes. (iii) We train the model to learn a joint representation of *disentangles* type, scale, and complexity, reflecting the orthogonal yet interacting aspects of MILP instances.

We evaluate GLIM on two MILP downstream tasks designed to probe embedding quality: (1) *MILP Instance Retrieval*, which retrieves similar instances for a given unseen query instance; and (2) *MILP Solver Hyperparameter Prediction*, which predicts solver hyperparameters for unseen instances. Across in-distribution, distribution-shifted settings, and standard benchmark MIPLIB (Gleixner et al., 2021), GLIM achieves strong performance and outperforms competitive baselines.

Our contributions can be summarized as follows:

- We introduce GLIM: a large-scale general-purpose MILP embedding model, trained on a diverse dataset spanning thousands of problem classes. Three key components of GLIM are the extended large-scale training dataset, the hybrid model architecture of GNN and Perceiver-like block, and the novel joint representation that disentangling type, scale, and complexity.

- We demonstrate that it is possible to learn an embedding model that generalizes across both data and tasks, while the model exhibits measurable generalization to out-of-distribution instances.

- We design and conduct comprehensive evaluation across two downstream tasks (instance retrieval and MILP Solver Hyperparameter Prediction) of MILP embedding model, demonstrating both effectiveness and generalization in real-world scenarios. These two downstream tasks also create benchmarks for future works.

## 2 RELATED WORKS

**Machine Learning for MILP** Our work falls into the category of ML-based approaches for MILPs. The predominant paradigm of ML-based solvers leverages neural networks to capture the structure of optimization problems, thereby learning heuristics that accelerate traditional solvers. Representative examples include predict-and-search (Han et al., 2023; Huang et al., 2024), learning to branch (Khalil et al., 2016; Labassi et al., 2022), learning to cut (Tang et al., 2020; Paulus et al., 2022), large neighborhood search (Wu et al., 2021; Ye et al., 2025) or even instance generation for data augmentation (Guo et al., 2024; Yang et al., 2024). Despite these advances, a key limitation is specialization: most models are tailored to a single problem class and task, leading to dramatic performance drops under distribution shift (Manchanda et al., 2022), and restricting transferability across tasks (e.g., a branching model cannot be applied to large neighborhood search). This naturally raises the question of whether a unified model can be trained across data distributions or tasks.

**Generalizable Learning Approaches for MILP** Early attempts at enhancing generalization in combinatorial optimization adopted diverse strategies. Some work introduced problem-specific adapters with a shared backbone to solve the problem (Drakulic et al., 2024), others convert multiclass combinatorial optimization problems into TSP problems and train a unified TSP solver (Pan

et al., 2025), while still others tokenized problem instances and solver trajectories to enable next-token prediction (Zong et al., 2025). Although these approaches demonstrated promising results, they remain limited to a small set of combinatorial problems and do not extend naturally to general MILPs. Further progress has been made with multi-task training, such as jointly learning tasks such as branching and predict-and-search (Cai et al., 2025), and with the use of large language models to synthesize large-scale MILP datasets, which were then used to train task-specific models for integral gap prediction, branching, and others (Li et al., 2025). Unlike existing work that is only applicable to a single task or a single class of problems, our work introduces a general-purpose MILP embedding model trained on a highly diverse dataset.

## 3 PRELIMINARY: MILP PROBLEM AND ITS DATA FORMS

The standard form of a Mixed-Integer Linear Programming (MILP) problem is:

$$
\begin{aligned}
\min_{x \in \mathbb{R}^n} \quad & c^\top x, \\
\text{s.t.} \quad & Ax \leq b, \\
& l \leq x \leq u, \\
& x_i \in \mathbb{Z}, \quad i \in \mathbb{I}.
\end{aligned}
\tag{1}
$$

In this formulation, the coefficient matrix $A \in \mathbb{R}^{m \times n}$ represents the constraints structure, $b \in \mathbb{R}^m$ denotes the constraints' right-hand side vector, and $c \in \mathbb{R}^n$ is the objective coefficient. Variable bounds are given by $l \in (\mathbb{R} \cup \{-\infty\})^n$ and $u \in (\mathbb{R} \cup \{+\infty\})^n$. The index set $\mathbb{I} \subseteq \{1, 2, \dots, n\}$ identifies variables restricted to integer values.

**Bipartite Graph Representation** The bipartite graph representation encodes MILP instances in a lossless manner (Gasse et al., 2019). Variables $\mathcal{V} = \{v_1, v_2, \dots, v_n\}$ and constraints $\mathcal{C} = \{c_1, c_2, \dots, c_m\}$ are modeled as disjoint node sets. An edge $e_{ij} = (v_i, c_j) \in \mathcal{E}$ is added whenever variable $v_i$ participates in constraint $c_j$. The resulting bipartite graph $\mathcal{G} = (\mathcal{V}, \mathcal{C}, \mathcal{E})$ captures structural relations between variables and constraints. Feature details are provided in Appendix B.1.

**Formulation Code** Formulation code provides a generative specification of MILP problem classes, written as Python programs using the PySCIPOpt library (Bolusani et al., 2024). Each code file encapsulates the procedural logic for constructing instances. Running the formulation code can directly generate the corresponding MILP instance.

The upper part of Figure 1 illustrates the GLIM pipeline for synthesizing training corpus starting from the formulation code. For examples of these data forms, please refer to Appendix A.1.3.

## 4 GLIM: PROPOSED APPROACH

In this section, we first present the key design of GLIM: disentangling MILP problems into separate representations of type, scale, and solving difficulty. We then introduce the corresponding model architecture, which incorporates Perceiver-style attention blocks together with tailored training objectives.

### 4.1 DISENTANGLED REPRESENTATION LEARNING OF MILP

Our central hypothesis is that the empirical difficulty and structural characteristics of a Mixed-Integer Linear Program (MILP) instance are driven by the interplay of three fundamental, quasi-orthogonal factors: instance type, instance scale, and solving complexity.

- **Instance Type** refers to the underlying combinatorial structure of the instance (e.g., Set Cover, Facility Location, Traveling Salesperson Problem). Instances of the same type share a common algebraic formulation and structural properties, regardless of their size.
- **Instance Scale** captures the dimensions of the instance, such as the number of variables, constraints, and non-zero coefficients. Scale directly influences the size of the search space.

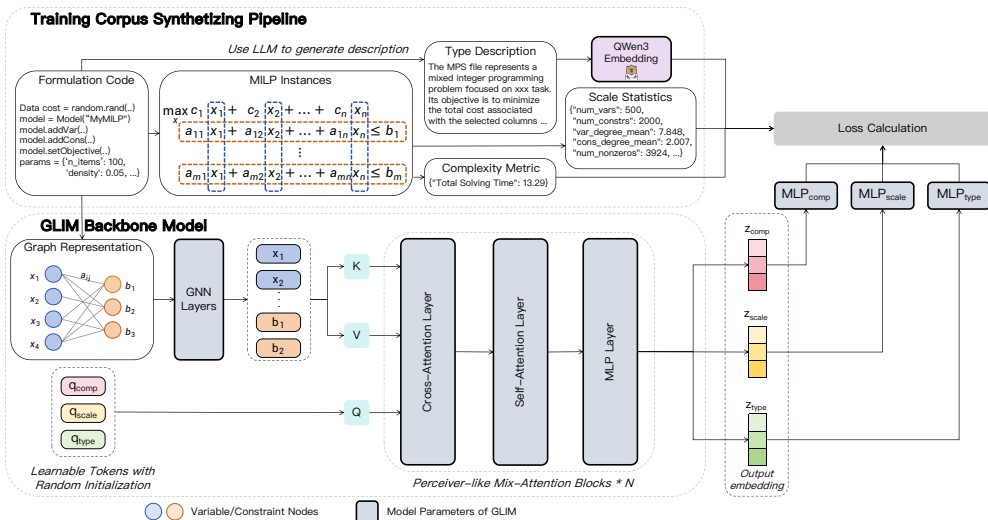

Figure 1: An overview of the proposed GLIM framework, which disentangles the representation of type, scale, and complexity of MILP instances. Regarding datasets, GLIM synthesizes the corresponding textual description, scale statics, and complexity metric for each instance as training labels. For model architecture, GLIM uses a GNN with Perceiver-like mix-attention blocks as a hybrid layer, making it adaptable to larger instance inputs.

- **Solving Complexity** is an emergent property reflecting the computational resources required by a solver to find the optimal solution. It is influenced by both type and scale but is not fully determined by them, as subtle structural variations can lead to dramatic differences in solving time.

Existing machine learning models for MILP often implicitly entangle these factors into a single, monolithic representation. This entanglement hinders generalization. For instance, a model trained on small-scale Set Cover instances may fail to recognize the "Set Cover" structure in a large-scale instance because the features related to scale dominate the representation.

To overcome this limitation, GLIM is designed to learn a *disentangled representation* that explicitly separates these three core factors. We define the final embedding $z$ of an MILP instance $I$ as the concatenation of three distinct sub-embeddings:

$$\text{GLIM}(I) = z = [z_{\text{type}}, z_{\text{scale}}, z_{\text{comp}}] \tag{2}$$

Each sub-embedding is trained to exclusively capture the information corresponding to its designated factor. $z_{\text{type}}$ aims to be invariant to changes in scale and complexity while encoding the problem's structural class. Conversely, $z_{\text{scale}}$ should capture dimensional statistics irrespective of the problem type. Finally, $z_{\text{comp}}$ serves as a proxy for the instance's intrinsic difficulty.

## 4.2 MODEL ARCHITECTURE

To effectively process large, variable-sized MILP instances and produce a disentangled representation, we design a hybrid architecture that combines a bipartite Graph Neural Network (GNN) encoder with a Perceiver-like readout mechanism.

**Bipartite GNN Encoder** Each MILP instance is first represented as a bipartite graph. The raw feature vectors for variable nodes, constraint nodes, and edges are projected into an embedding space using separate linear layers. The core of the encoder consists of several layers of bipartite graph convolution Kipf (2016). Specifically, an updated constraint embedding is computed by aggregating features from its neighboring variable nodes and the corresponding edges, and vice versa for variable embeddings. This GNN encoder produces a set of context-aware node-level embeddings for every variable and constraint in the MILP instance.

**Perceiver-like Mix-Attention Block**    A key challenge is to aggregate the variable number of node embeddings (which can be in the tens of thousands) into a fixed-size, disentangled graph-level representation. Simple global pooling methods risk losing critical information. Inspired by the Perceiver architecture (Jaegle et al., 2021), we introduce an attention-based readout mechanism to distill information into our three target factors.

We first initialize three learnable latent vectors, which we term the *type token* $q_{\text{type}}$, *scale token* $q_{\text{scale}}$ and *complexity token* $q_{\text{comp}}$. These tokens act as queries. The full set of variable and constraint embeddings produced by the GNN encoder serves as the key and value context. The three latent tokens then perform cross-attention over all node embeddings in the graph. This allows each token to selectively aggregate information relevant to its specific purpose. For instance, the type token learns to focus on structural patterns indicative of the problem class, while the scale token focuses on features related to the graph's size.

This cross-attention block is followed by a self-attention layer over the three latent tokens and a feed-forward network, allowing the distilled representations to be further refined. This entire Perceiver block can be stacked for multiple iterations. To handle extremely large graphs, we incorporate a top-K selection mechanism on the node embeddings based on their L2-norm before the attention step, ensuring computational tractability. The output of this stage is three distinct latent embeddings, $z_{\text{type}}$, $z_{\text{scale}}$ and $z_{\text{comp}}$, which form our disentangled representation.

**Projection Heads**    Finally, the three disentangled latent embeddings are passed to separate Multi-Layer Perceptron (MLP) heads. Each head is tailored to a specific prediction task aligned with our training objectives. The type head projects $z_{\text{type}}$ into an embedding space for contrastive learning, while the scale and complexity heads regress from $z_{\text{scale}}$ and $z_{\text{comp}}$ to their respective target values.

### 4.3 TRAINING OBJECTIVES

To encourage disentanglement, GLIM is trained with a composite multi-task loss that jointly supervises *type*, *scale*, and *complexity* factors:

$$\mathcal{L} = \lambda_{\text{type}}\mathcal{L}_{\text{type}} + \lambda_{\text{scale}}\mathcal{L}_{\text{scale}} + \lambda_{\text{comp}}\mathcal{L}_{\text{comp}}.$$

Here, $\lambda_{\text{type}}, \lambda_{\text{scale}}, \lambda_{\text{comp}}$ balance the contributions of the three components. Each sub-objective supervises a distinct embedding head:

- The *type loss* adopts a contrastive formulation similar to CLIP (Radford et al., 2021), aligning each MILP instance with its textual formulation while pushing apart mismatched pairs.
- The *scale loss* is a regression task predicting structural statistics (e.g., number of variables and constraints) from the scale embedding, encouraging sensitivity to problem size.
- The *complexity loss* regresses the solver time from the complexity embedding, providing a signal that captures computational hardness.

Together, these objectives force the model to encode complementary and interpretable factors within its embedding. Full mathematical definitions are provided in Appendix B.2.

## 5 DATASETS

For an embedding model, the diversity and quality of training data are critical. In this section, we describe how we construct the dataset used by GLIM and the corresponding training labels.

### 5.1 IMPROVING DATA QUALITY

**Improving Scale Diversity**    We begin with the largest publicly available dataset of MILPs (Li et al., 2025), which provides formulation codes for 2,000 problem classes. While this dataset offers rich problem-type diversity, each formulation typically generates instances at a fixed scale. To enrich scale diversity, we prompt LLM with each formulation code to produce multiple parameter sets, which we then inject back into the code to obtain new formulation variants capable of generating larger or smaller instances. Finally, by varying random seeds within each formulation, we obtain a wide set of distinct MILP instances.

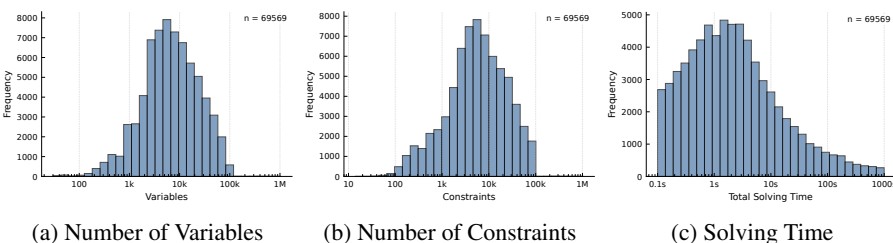

(a) Number of Variables      (b) Number of Constraints      (c) Solving Time

Figure 2: Distribution of training data used by GLIM.

**Data Filtering** The procedure above inevitably produces some infeasible or unbounded instances. To address this, we solve every candidate instance and filter out those that are infeasible or unbounded. We further prune the dataset to remove instances with (i) solving time is greater than 1000 seconds or less than 0.1 seconds, (ii) the sum of variables and constraints larger than 100,000, or (iii) more than 150,000 nonzero entries in the coefficient matrix. After filtering, we obtain a high-quality dataset containing 77,286 MILP instances.

## 5.2 DATA LABEL CONSTRUCTION

To align with GLIM's objective of disentangling type, scale, and complexity in its embeddings, we construct labels along these three dimensions: **(1) Type label**. For each instance, we input its formulation code into an LLM and explicitly prompt for descriptive characteristics. We then pass this textual description through a text embedding model, and the resulting vector is used as the type label. **(2) Scale label**. We extract eight standard statistics (e.g., number of variables) as the scale label. **(3) Complexity label**. We directly use the solving time of the instance as its complexity label. In order to unify the solving environment, all solving times in the paper are obtained by Gurobi 12.0.2 (Gurobi Optimization, LLC, 2024), which is limited to single-threaded solving.

Since both instance statistics and solving times are heavy-tailed, we take the logarithm of raw values for scale and complexity labels. This log transformation better aligns with empirical distributions and stabilizes training.

## 5.3 DATASET SPLITS

From the filtered dataset, we randomly split 90% for training and 10% for validation. The validation set serves as *in-distribution* data for evaluating GLIM under matched conditions, namely "GLIM-valid". To further probe *out-of-distribution* generalization, we construct two additional datasets. The first one includes 4 benchmark problems (Multi-Item Lot Sizing, Graph Coloring, Bin Packing, Job Scheduling) that have not appeared in the training data. The second one is a dataset derived from the MIPLIB collection set (Gleixner et al., 2021), where we apply the same filtering rules. Out of 1,065 instances, 413 remain after filtering and are used for testing.

We visualize the distribution of the number of variables/constraints and solution time for the training split in Figure 2. For detailed description and statistics, please refer to Appendix A.1.

## 6 EVALUATION

Our experiments aim to address the following research questions: (i) Can GLIM effectively represent MILP instances belonging to the problem types seen during training? (ii) How well does GLIM perform when applied to unseen or out-of-distribution MILP instances? (iii) To what extent do the scale of the training data and model architecture contribute to GLIM's overall performance? We begin by describing the experimental setup, including the baselines and evaluation tasks.

**Baselines** To the best of our knowledge, no general-purpose MILP embedding models exist to serve as direct baselines. Consequently, we structure our comparative analysis in two parts. First, we conduct a series of ablation studies to isolate the contributions of GLIM's key components. Specifically, we used data of different sizes to train GLIM-S, GLIM-M, and GLIM-L. We also conducted ablation experiments on GLIM's Perceiver-like attention block (results are in Appendix D.1). Sec-

Table 1: MEE (Mean Excess Error) results of instance retrieval, with retrieval target set as **problem scale**. Lower is better.

| Model | In-Distribution | Out-Of-Distribution | | | | |
|---|---|---|---|---|---|---|
| | GLIM-valid | Multi-Item Lot Sizing | Graph Coloring | Bin Packing | Job Scheduling | MIPLIB |
| Linq-Embed-Mistral | 0.093 | 0.871 | 3.304 | 3.224 | 2.215 | 1.601 |
| Qwen-3-Embed | 0.096 | 1.534 | 1.409 | 1.706 | 1.316 | 1.681 |
| GLIM-S | 0.038 | 2.314 | 1.397 | 0.898 | 1.491 | 1.442 |
| GLIM-M | 0.026 | 0.759 | **1.021** | 0.575 | 2.055 | 1.255 |
| GLIM-L | **0.024** | **0.315** | 1.151 | **0.561** | **1.194** | **1.203** |

Table 2: MEE (Mean Excess Error) results of instance retrieval, with retrieval target set as **problem complexity**. Lower is better.

| Model | In-Distribution | Out-Of-Distribution | | | | |
|---|---|---|---|---|---|---|
| | GLIM-valid | Multi-Item Lot Sizing | Graph Coloring | Bin Packing | Job Scheduling | MIPLIB |
| Linq-Embed-Mistral | 0.914 | 4.801 | 3.170 | **1.886** | 2.295 | 3.474 |
| Qwen-3-Embed | 0.904 | 3.603 | 2.241 | 3.072 | 4.643 | 3.483 |
| GLIM-S | 0.671 | 2.485 | 3.399 | 3.503 | 2.815 | 4.053 |
| GLIM-M | 0.678 | 2.266 | 2.508 | 3.389 | 2.603 | 3.494 |
| GLIM-L | **0.645** | **2.077** | **2.180** | 2.846 | **1.678** | **3.357** |

ond, we establish baselines using SOTA text embedding models Qwen-3-Embedding (Zhang et al., 2025) and Linq-Mistral-Embed (Choi et al., 2024). For this, we convert each MILP instance into its human-readable textual format and encode this representation. Detailed implementation of all baselines is provided in Appendix A.2.

**Evaluation Tasks**  Unlike established domains like text (Muennighoff et al., 2022) or image (Xiao et al., 2025), which benefit from standard embedding model benchmarks, the evaluation of general-purpose MILP representations requires the design of suitable downstream tasks. To this end, we propose two practical tasks **MILP Instance Retrieval** and **MILP Solver Hyperparameter Prediction** to probe the quality and utility of the learned embeddings. We evaluate the generalization ability of the GLIM model by testing on out-of-distribution data.

## 6.1    Task: MILP Instance Retrieval

To assess the quality of the learned embeddings for instance retrieval, we evaluate their ability to identify instances with similar characteristics from a large library, which is analogous to retrieval tasks in other modalities. This task is crucial as it directly tests the model's capacity to encode salient features into a discriminative latent space. We define similarity based on two practical target attributes: scale-based features (e.g., number of variables) and complexity-based features (e.g., solution time), which are critical for applications like data augmentation for ML-based solvers and heuristic generation.

### 6.1.1    Instance Retrieval Accuracy

For a given query instance, we embed it into a token representation corresponding to the chosen attribute (either scale or complexity). We then compute the cosine similarity between this query token and the corresponding tokens of all instances in a predefined library, and the instance with the highest similarity is retrieved.

Here we define an indicator **Excess Error**, which indicates how much the error between the current retrieved question and the target question is worse than the minimum error between the target question and the retrieval database (i.e., the ideal retrieval result). The retrieval quality is quantitatively measured using the **Mean Excess Error (MEE)**, which averages the instance-level Excess Error. For a single query, Excess Error is defined as:

$$\text{EE} = \sum_{i=0}^{N-1} \left| \log(y_{\text{retrieved,i}}) - \log(y_{\text{true,i}}) \right|, \tag{3}$$

where $y_{\text{retrieved,i}}$ is the i-th attribute value of the retrieved instance and $y_{\text{true}}$ is the i-th attribute value of the most similar instance in the library under the target metric, and $N$ is the number of attribute

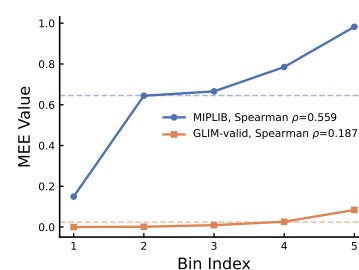

Figure 3: Results on instance retrieval task, target=scale.

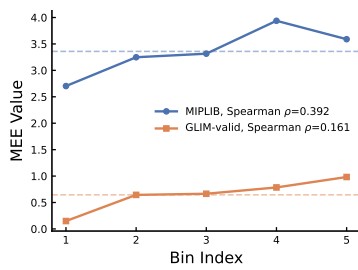

Figure 4: Results on instance retrieval task, target=complexity.

values. The final result MEE is calculated by averaging the EE values over all instances in the evaluation set. This metric effectively quantifies the discrepancy in the target numerical attribute between the retrieved instances and the ideal match, providing a robust measure of retrieval precision.

Leveraging the disentangled embeddings produced by GLIM, we can evaluate retrieval tasks based on different instance attributes, specifically by scale and by complexity. For scale-targeted retrieval, we use the output embedding token $z_{\text{scale}}$ to retrieve instances. The attribute values used for evaluation include the instance's number of variables, constraints, and non-zero elements in coefficient matrix. For complexity-targeted retrieval, we utilize the output embedding token $z_{\text{comp}}$ to retrieve instances. The instance's solution time is used as the attribute value for evaluation.

Tables 1 and 2 present the results of retrieval based on scale and complexity, respectively. We fixed the GLIM training dataset as the retrieval library and tested it on various datasets. We can see that GLIM outperforms the baseline on in-distribution datasets, unseen standard problems, and even MIPLIB in most cases. Furthermore, GLIM trained on larger datasets achieves even better performance, demonstrating the effectiveness of increasing the size of its training data.

### 6.1.2 PROBING THE GENERALIZATION ABILITY OF THE MODEL

Our evaluation so far has omitted the $z_{\text{type}}$ embedding because out-of-distribution test sets such as MIPLIB lack ground-truth type labels. To address this, we assess $z_{\text{type}}$ from a different perspective: as a measure of an instance's deviation from the training distribution. The intuition is that for an unseen instance, a larger distance in the $z_{\text{type}}$ space indicates greater dissimilarity from the training data. Formally, we define the deviation score for an instance $k$ as $d_k = \max_{c \in \mathcal{S}} z_{\text{type},k} \cdot z_{\text{type},c}$, where $\mathcal{S}$ denotes the set of training instances used by GLIM, and $z_{\text{type},k}$ is the type embedding token obtained by encoding instance $k$ with GLIM. A higher $d_k$ indicates greater similarity between the type of instance $k$ and the training distribution of GLIM.

We evaluate the correlation between the deviation score of each test instance and its Excess Error (EE) value. Specifically, we partition the instances in each dataset into five equally sized groups based on their deviation scores and compute the mean EE value within each group. In addition, we compute the Spearman correlation coefficient between the deviation scores and the EE values. Experimental results targeting instance scale and complexity are reported in Table 3 and Table 4, respectively. These results quantify the generalization performance of our GLIM model.

### 6.2 TASK: MILP SOLVER HYPERPARAMETER PREDICTION

It is well-known that solver hyperparameters can significantly impact the solution time for MILP instances. Therefore, a key research objective is to directly predict a near-optimal set of solver hyperparameters for a given unseen MILP instance to accelerate its solving process. This task evaluates the embedding's ability to generalize to a practical, unseen task: predicting effective solver hyperparameter.

### 6.2.1 EVALUATION PROTOCOL

We begin by constructing a dataset where each input is a MILP instance and each label corresponds to a near-optimal set of solver hyperparameters for solving that instance. Using this dataset, we train

Table 3: Win rate results for the *MILP Solver Hyperparameter Prediction* task.

| Model | Out-Of-Distribution | | | | |
|---|---|---|---|---|---|
| | Multi-Item Lot Sizing | Graph Coloring | Bin Packing | Job Scheduling | MIPLIB |
| Linq-Embed-Mistral | 30/49 | 23/28 | 40/45 | 45/50 | 196/413 |
| Qwen-3-Embed | 45/49 | 27/28 | 28/45 | 42/50 | 153/413 |
| GLIM-S | 46/49 | 18/28 | 43/45 | 30/50 | 175/413 |
| GLIM-M | 21/49 | 26/28 | 45/45 | 47/50 | 210/413 |
| GLIM-L | 31/49 | 24/28 | 40/45 | 49/50 | 199/413 |

Table 4: Average moderated solving time results for the *MILP Solver Hyperparameter Prediction* task. The ratio in parentheses is the average speedup ratio.

| Model | Out-Of-Distribution | | | | |
|---|---|---|---|---|---|
| | Multi-Item Lot Sizing | Graph Coloring | Bin Packing | Job Scheduling | MIPLIB |
| Default | 1.073 | 85.020 | 2.306 | 4.791 | 91.155 |
| Linq-Embed-Mistral | 1.057 (1.53) | 25.954 (10.15) | 0.808 (2.76) | 3.496 (5.04) | 78.438 (4.19) |
| Qwen-3-Embed | 0.855 (2.69) | 14.189 (6.96) | 2.029 (1.78) | 3.593 (5.83) | 81.956 (4.36) |
| GLIM-S | 0.679 (2.40) | 74.039 (4.75) | 0.678 (3.41) | 4.338 (2.49) | 80.292 (4.19) |
| GLIM-M | 1.054 (1.71) | 16.557 (7.28) | 0.475 (5.52) | 3.744 (4.84) | 73.570 (4.84) |
| GLIM-L | 1.053 (1.83) | 17.862 (7.82) | 1.133 (2.46) | 0.761 (8.62) | 73.161 (5.46) |

a lightweight MLP head on top of the GLIM backbone. The MLP takes the concatenated embedding $[z_{\text{type}}, z_{\text{scale}}, z_{\text{comp}}]$ of a MILP instance as input and predicts the associated hyperparameter settings.

For this task, we evaluate different variants of GLIM as well as text embedding models as baselines. All instances are solved using Gurobi with a single-thread setting. The evaluation metric is the solving time, compared against the default hyperparameter. For more details about the evaluation protocol, please refer to Appendix C.

### 6.2.2 RESULTS IN SOLVER HYPERPARAMETER PREDICTION

We evaluate on four out-of-distribution benchmark classes as well as MIPLIB to assess performance. We firstly report **Win Rate**, defined as the proportion of instances where solving with the predicted hyperparameter achieves better runtime than using the solver's default hyperparameter. Results are summarized in Table 3, showing that GLIM variants predict hyperparameter superior to the default hyperparameter for most problems across both the four benchmarks and MIPLIB.

It is important to note that in some cases, the predicted hyperparameter achieves a high win rate to default hyperparameter, but yield worse average solving time. This occurs because in a small fraction of instances, the MLP head predicts suboptimal hyperparameter that degrades performance, inflating the average runtime. So we report results under a **Moderated Solving Time** metric, defined as the minimum between the runtime with predicted hyperparameter and that with the default hyperparameter. We then report the average moderated runtime as well as the corresponding average **speedup ratio**. Results in Table 4 provide strong evidence for the practical effectiveness of GLIM on out-of-distribution problems. For experimental results on the in-distribution dataset, see Appendix D.2.

## 7 CONCLUSION

In this paper, we proposed GLIM, a general-purpose embedding model for MILPs. We demonstrated that it is possible to train a shared backbone model across 2,000 classes of MILP problems. Beyond achieving strong performance on in-distribution data, GLIM exhibits robust generalization to unseen, out-of-distribution datasets. Our results highlight two key benefits: (i) as an MILP embedding model, GLIM effectively captures problem type, scale, and solving complexity; and (ii) as a general-purpose backbone, it can be readily adapted to downstream task. While a performance gap remains between in-distribution and out-of-distribution settings, GLIM offers a particularly promising step toward developing unified models for combinatorial optimization. To the best of our knowledge, this represents one of the first attempts to train a single model that generalizes across diverse distributions and tasks without fine-tuning on unseen problems. Looking forward, an important direction is to further narrow the gap between in-distribution and out-of-distribution performance, enabling even stronger transfer across problem domains.

## ETHICS STATEMENT

The methods proposed in this paper aim to learn a unified representation for diverse Mixed-Integer Linear Programs, which is related to the broader field of neural combinatorial optimization and representation learning. To our best knowledge, no ethical issues or harmful insights of this work need to be otherwise stated.

## REPRODUCIBILITY STATEMENT

The datasets used and the baseline implementation are described in Appendix A. The detailed hypermeters and implementation of the models for training and testing are provided in Appendix B. Source code and datasets can be accessed at `https://anonymous.4open.science/r/GLIM-DBAF`.

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

TABLE OF CONTENTS FOR APPENDIX

# A DETAILS ON EXPERIMENTS

## A.1 DATASETS

### A.1.1 CONSTRUCTION OF DATASETS

We first obtain the formulation codes for 2,000 classes of MILP problems from `https://huggingface.co/datasets/microsoft/MILP-Evolve/`, which are already sufficiently diverse in type. However, each formulation code can only generate problems of a fixed scale. We observe that the parameter set within each formulation code can be adjusted to produce problems of varying scales. Therefore, we prompt a large language model (LLM) with each formulation code to generate multiple parameter sets. Specifically, we employ GPT-4o (Hurst et al., 2024) as the LLM with the following prompt:

---

**Prompt to diversify instance scale**

You are a world-class expert in optimization and algorithmic problem solving. Your task is to generate instances of the same class of problems with varying scales or levels of difficulty.

In the provided code, a MILP (Mixed-Integer Linear Programming) problem is defined, generated, and solved. In the 'main' function of the code, there is a Python Dict named 'parameters' that describes the input parameters of the problem.

You should carefully analyze the problem context and provide a new 'parameters' Dict in JSON format that results in a problem of a different scale when used with the code. The new instance should differ meaningfully from the current one, but not be too easy (solving time $\leq$ 1s) or too hard (solving time $\geq$ 100s), and the new instance can be larger or smaller than the original one.

---

For each original formulation code, we generate 10 different parameter sets, resulting in a total of 20,000 formulation codes. From each formulation code, we further generate problems using 5 different random seeds, yielding 100,000 problem instances in total. Not all of these instances are guaranteed to be feasible, and some may be unsolvable or too large/small in scale or difficulty. Following the filtering procedure described in Section 5.1, we ultimately obtain 77,295 valid problem instances. These are randomly split into GLIM-train and GLIM-valid with a 9:1 ratio.

For testing on in-distribution data, we consider two tasks. For the first task, MILP instance retrieval, we evaluate using the entire GLIM-valid dataset. For the second task, MILP solver hyperparameter prediction, we select four benchmark problem classes (Combinatorial Auction Problem, Generalized Independent Set Problem, Fixed Charge Multi-Commodity Network Flow Problem, Set Cover Problem) that appear in the training set, and generate additional instances for evaluation.

For testing on out-of-distribution data, we evaluate on four standard problem classes: Multi-Item Lot Sizing, Graph Coloring, Bin Packing, and Job Scheduling, as well as the MIPLIB Collection Set, none of which appear in the training data.

### A.1.2 DETAILS OF DATA LABELS

In the training process of GLIM, we employ three types of data labels: the *type label*, the *scale label*, and the *complexity label*.

**Type label** The type label captures the categorical information of a problem instance. Since all MILP instances used in GLIM training are synthetically generated from MILP formulation codes, the formulation code itself fully encodes the problem type. To extract this information, we input the formulation code into an LLM and design a prompt that instructs the LLM to produce a textual description of the corresponding MILP instance:

---

**Prompt to generate type description**

You are a world-class expert in optimization and algorithmic modeling.

You are given a Python code that defines, generates, and solves a Mixed-Integer Linear Programming (MILP) problem. Your task is to write a single, high-quality paragraph that provides a **detailed, comprehensive, and scale-independent** description of the MILP problem defined by the code. This description should not rely on any specific parameter values (e.g. from the parameters dict or instance sizes), but instead focus on the **general formulation and structure** of the problem.

Your response must consist of a single, complete paragraph, and should focus on (but not be limited to) the following key aspects:
- The problem domain (e.g. network design, resource allocation),
- The formulation used (including how flows, capacities, and decisions are modeled),
- The different types of decision variables (continuous, integer, binary) and what they represent,
- The constraints (e.g. flow conservation, capacity constraints, budget limits, worker-resource coupling),
- The objective function, including all cost components and trade-offs being optimized.

Avoid using any dataset-specific values. Focus instead on the logic, relationships, and mathematical modeling choices in the code. Write your response as if explaining the essence of the MILP formulation to a fellow expert who has not seen the code but is capable of understanding advanced optimization models.

---

We then feed the generated textual description into the text embedding model `Qwen-3-embedding`, obtaining a type label representation $l_{\text{type}} \in \mathbb{R}^{4096}$. All MILP instances belonging to the same category share the same type label.

**Scale label and complexity label** The scale label encodes the size-related characteristics of an MILP instance. Specifically, we select eight core statistical features, summarized in Table 5. The complexity label corresponds to the solving time of the instance. We compute this by solving the MILP instance with the Gurobi 12.0.2 solver (Gurobi Optimization, LLC, 2024), restricted to a single CPU core to ensure a uniform computational environment.

Table 5: MILP statistical indicators used for scale label.

| Indicator | Description |
|---|---|
| num_vars | Number of variables |
| num_constrs | Number of constraints |
| num_nonzeros | Number of nonzero coefficients in the constraint matrix |
| coef_dens | Density of coefficients |
| cons_degree_mean | Average degree of constraints |
| cons_degree_std | Standard deviation of constraint degrees |
| var_degree_mean | Average degree of variables |
| var_degree_std | Standard deviation of variable degrees |

### A.1.3 SAMPLES OF DIFFERENT FORMS OF MILP DATA

Here we provide a sample of code and corresponding textual description in training dataset. Lines 91-98 in the code correspond to the parameter part of the code, which we change to generate problems of different sizes.

**Formulation Code**

```python
1  import random
2  import time
3  import numpy as np
4  import networkx as nx
5  from pyscipopt import Model, quicksum
6
7  class GISP:
8      def __init__(self, parameters, seed=None):
9          for key, value in parameters.items():
10             setattr(self, key, value)
11
12         self.seed = seed
13         if self.seed:
14             random.seed(seed)
15             np.random.seed(seed)
16
17     ################# Data Generation #################
18     def generate_random_graph(self):
19         n_nodes = np.random.randint(self.min_n, self.max_n)
20         G = nx.erdos_renyi_graph(n=n_nodes, p=self.er_prob, seed
               =self.seed)
21         return G
22
23     def generate_revenues_costs(self, G):
24         if self.set_type == 'SET1':
25             for node in G.nodes:
26                 G.nodes[node]['revenue'] = np.random.randint(1,
                      100)
27             for u, v in G.edges:
28                 G[u][v]['cost'] = (G.nodes[u]['revenue'] + G.
                      nodes[v]['revenue']) / float(self.set_param)
29         elif self.set_type == 'SET2':
30             for node in G.nodes:
31                 G.nodes[node]['revenue'] = float(self.set_param)
32             for u, v in G.edges:
33                 G[u][v]['cost'] = 1.0
34
35     def generate_removable_edges(self, G):
36         E2 = set()
37         for edge in G.edges:
38             if np.random.random() <= self.alpha:
39                 E2.add(edge)
40         return E2
41
42     def generate_instance(self):
43         G = self.generate_random_graph()
44         self.generate_revenues_costs(G)
45         E2 = self.generate_removable_edges(G)
46         res = {'G': G, 'E2': E2}
47
48         return res
49
50     ################# PySCIPOpt Modeling #################
51     def solve(self, instance):
52         G, E2 = instance['G'], instance['E2']
53
54         model = Model("GISP")
55         node_vars = {f"x{node}":  model.addVar(vtype="B", name=f
               "x{node}") for node in G.nodes}
```

```python
56          edge_vars = {f"y{u}_{v}": model.addVar(vtype="B", name=f
                "y{u}_{v}") for u, v in G.edges}
57
58          objective_expr = quicksum(
59              G.nodes[node]['revenue'] * node_vars[f"x{node}"]
60              for node in G.nodes
61          )
62
63          objective_expr -= quicksum(
64              G[u][v]['cost'] * edge_vars[f"y{u}_{v}"]
65              for u, v in E2
66          )
67
68          for u, v in G.edges:
69              if (u, v) in E2:
70                  model.addCons(
71                      node_vars[f"x{u}"] + node_vars[f"x{v}"] -
                          edge_vars[f"y{u}_{v}"] <= 1,
72                      name=f"C_{u}_{v}"
73                  )
74              else:
75                  model.addCons(
76                      node_vars[f"x{u}"] + node_vars[f"x{v}"] <=
                          1,
77                      name=f"C_{u}_{v}"
78                  )
79
80          model.setObjective(objective_expr, "maximize")
81
82          start_time = time.time()
83          model.optimize()
84          end_time = time.time()
85
86          return model.getStatus(), end_time - start_time
87
88
89  if __name__ == '__main__':
90      seed = 42
91      parameters = {
92          'min_n': 70,
93          'max_n': 100,
94          'er_prob': 0.6,
95          'set_type': 'SET2',
96          'set_param': 100.0,
97          'alpha': 0.5
98      }
99
100     gisp = GISP(parameters, seed=seed)
101     instance = gisp.generate_instance()
102     solve_status, solve_time = gisp.solve(instance)
103
104     print(f"Solve Status: {solve_status}")
105     print(f"Solve Time: {solve_time:.2f} seconds")
```

**Textual Description**

The Generalized Independent Set Problem (GISP) is a combinatorial optimization problem that extends the classical Independent Set Problem by incorporating edge removal decisions with associated costs, formulated as a Mixed-Integer Linear Program (MILP). The problem operates on an undirected graph where nodes represent entities with associated revenues, and edges represent conflicts or dependencies between these entities. The objective is to select a subset of nodes that maximizes the total revenue of the selected nodes while accounting for the costs of removing certain edges to resolve conflicts, subject to constraints ensuring the selected nodes form an independent set (i.e., no two selected nodes are adjacent). The decision variables include binary node-selection variables indicating whether a node is included in the independent set and binary edge-removal variables for a subset of removable edges, which allow the relaxation of conflict constraints at a cost. The constraints enforce that either at most one node from any pair of adjacent nodes is selected or the edge between them is removed (if removable), ensuring the solution's feasibility. The objective function balances the trade-off between maximizing the total revenue from selected nodes and minimizing the total cost of edge removals, with the edge costs being either derived from node revenues or set to a constant, depending on the problem configuration. This formulation captures the interplay between node selection and edge removal, making it suitable for applications requiring conflict resolution or resource allocation under interdependencies.

## A.2 BASELINES

**Text Embedding Models** Since there are no existing works on MILP embedding models to serve as baselines, we consider a broader alternative for comparison: text embedding models. MILP problems in .lp or .mps format are represented as human-readable text, which makes them suitable for input into a text embedding model. However, the context window limits of current text embedding models are relatively small (e.g., 4096 or 8192 tokens). As a result, for large-scale problems, the entire textual representation cannot be directly fed into the model, and only a sampled subset can be used. Following the sampling strategy described in (cite), we extract 150 lines of information from each instance: the first 50 lines, 50 lines sampled at random from the middle, and the last 50 lines. An example of the sampled information from one instance is shown below:

**Example of sampled instance**

```
* SCIP STATISTICS * Problem name : GISP
* Variables : 2350 (2350 binary, 0 integer, 0 implicit integer, 0 continuous)
* Constraints : 2242
NAME GISP
OBJSENSE MIN
ROWS N Obj L C_0_4 L C_0_8 L C_0_13 L C_0_16
(omitted...)
COLUMNS
INTSTART 'MARKER' 'INTORG'
x0 Obj -100 C_0_68 1
x0 C_0_70 1 C_0_71 1
x0 C_0_72 1 C_0_76 1
x0 C_0_84 1 C_0_86 1
x0 C_0_98 1 C_0_103 1
(omitted...)
BV Bound y103_105
BV Bound y103_106
BV Bound y104_107
ENDATA
```

We adopt the current state-of-the-art text embedding models `Qwen-3-embedding-8b` and `Linq-mistral-embed` for comparison. To align with GLIM's disentangled embeddings in the three dimensions of *type*, *scale*, and *complexity*, we design instruction prompts for the text embedding models. This enables them to also produce three sub-embeddings, $[z_{\text{type}}, z_{\text{scale}}, z_{\text{comp}}]$. We further set the output dimensionality of the text embedding models to 256, consistent with GLIM's design. The instruction prompts for extracting the type/scale/complexity embeddings are as follows:

---

**Instruction prompt for text embedding model**

(For type embedding) Given a Mixed Integer Linear Programming (MILP) problem instance in MPS format below, retrieve a similar problem instance that belongs to the same problem category, domain application, or mathematical structure type.

(For scale embedding) Given a Mixed Integer Linear Programming (MILP) problem instance in MPS format below, retrieve a similar problem instance that has matching statistical properties including number of variables, constraints, non-zero elements, and coefficient distributions.

(For complexity embedding) Given a Mixed Integer Linear Programming (MILP) problem instance in MPS format below, retrieve a similar problem instance that has comparable computational complexity, solution difficulty, and optimization challenge level.

---

**GLIM Variants**  We trained three different versions of the GLIM model (GLIM-S, GLIM-M, and GLIM-L) for comparison. These models share the same architecture but differ in the amount of training data used. The training data scales are summarized in Table 6. In addition, Figures 9–12 visualize the loss curves of these three models on `GLIM-valid` during training.

Table 6: Training data statistics for GLIM-S, GLIM-M, and GLIM-L.

| Model | Num. of MILP Classes | Num. of MILP Instances |
|---|---|---|
| GLIM-S | 50 | 1,908 |
| GLIM-M | 400 | 13,925 |
| GLIM-L | 2000 | 69,569 |

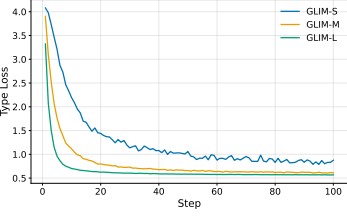

Figure 5: Loss curve of $\mathcal{L}_{\text{type}}$.

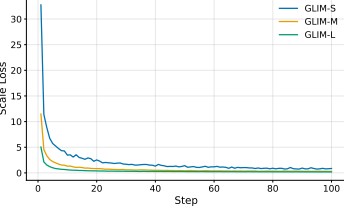

Figure 6: Loss curve of $\mathcal{L}_{\text{scale}}$.

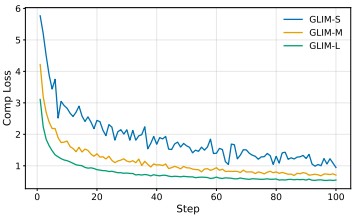

Figure 7: Loss curve of $\mathcal{L}_{\text{comp}}$.

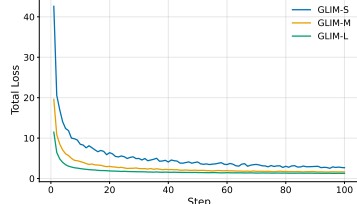

Figure 8: Total Loss curve.

# B IMPLEMENTATION DETAILS OF GLIM

## B.1 DETAILS OF BIPARTITE GRAPH FEATURES

To encode an MILP instance as a corresponding bipartite graph, we incorporate information about both variables and constraints into the node features of the graph representation. The specific node features used in our encoding are detailed in Table 7.

Table 7: Node type features and descriptions for Variables and Constraints.

| Node Type | Feature | Description |
|---|---|---|
| **Vars** | type | Variable type |
| | lb | Variable lower bound |
| | ub | Variable upper bound |
| | norm_coef | Objective coefficient normalized by objective norm |
| **Cons** | row_norm | $\ell_2$-norm of row coefficients |
| | obj_norm | $\ell_2$-norm of objective coefficients |
| | lhs | Left-hand side (depending on constraint sense) |
| | rhs | Right-hand side (depending on constraint sense) |
| | ncols | Total number of variables in the model |
| | nlpnonz | Number of nonzero coefficients in the row |
| | cst | Row constant term |
| | origin_type | Constraint sense: $\{<, =, >, \text{other}\}$ |
| | intcols | Count of integral coefficients in the row |
| | rank | Normalized row index |

In each layer, messages are passed iteratively between variable and constraint nodes, allowing their representations to be mutually refined based on the graph's structure.

We employ residual connections across GNN layers and apply GraphNorm (Cai et al., 2021) to the final node embeddings to stabilize training and improve performance on graphs of varying sizes.

## B.2 DETAILS OF MODEL ARCHITECTURE

**Feature Embedding** We embed rows/cols with two-layer MLPs and normalize edges:

$$h_i^{r,0} = \phi_r(x_i^r) = \text{GELU}(W_{r,2}\,\text{Drop}(\text{GELU}(W_{r,1}\,\text{Norm}(x_i^r)))), \tag{4}$$

$$h_j^{c,0} = \phi_c(x_j^c) = \text{GELU}\big(W_{c,2}\,\text{Drop}\big(\text{GELU}\big(W_{c,1}\,\text{Norm}(x_j^c)\big)\big)\big), \tag{5}$$

$$\tilde{e}_{ij} = \text{Norm}(e_{ij}), \tag{6}$$

where Norm is either LayerNorm, and Drop denotes dropout.

**Bipartite Message Passing Encoder** We employ alternating bipartite convolutions for $T$ iterations. One layer from columns to rows reads

$$m_i^{r,t} = \sum_{j \in \mathcal{N}(i)} \psi\big(h_i^{r,t}, h_j^{c,t}, \tilde{e}_{ji}\big), \quad \psi(a,b,c) = \text{LN}\big(W_\ell a + W_e c + W_r b\big), \tag{7}$$

$$h_i^{r,t+1} = \underbrace{U_r\big([m_i^{r,t}; h_i^{r,t}]\big)}_{\text{post-conv \& fusion}}, \qquad U_r(z) = W_{o,r}\,\text{GELU}(z), \tag{8}$$

and symmetrically a layer from rows to columns:

$$m_j^{c,t} = \sum_{i \in \mathcal{N}(j)} \psi\left(h_j^{c,t}, h_i^{r,t+1}, \tilde{e}_{ij}\right), \tag{9}$$

$$h_j^{c,t+1} = U_c\big([m_j^{c,t}; h_j^{c,t}]\big). \tag{10}$$

When $T > 1$, residual accumulation is applied across iterations (i.e., $h^{\cdot,t+1} \leftarrow h^{\cdot,t} + (\cdot)$). Optionally, an edge-aware multi-head GAT variant can be used in place of $\psi$. After message passing we apply GraphNorm per part:

$$h^r = \text{GraphNorm}\big(\text{GELU}(W_{r,3} h^{r,T})\big), \quad h^c = \text{GraphNorm}\big(\text{GELU}(W_{c,3} h^{c,T})\big). \quad (11)$$

**Token Construction and Side Embeddings**    Let $H_r = \{h_i^r\}_{i \in \mathcal{V}_r}$ and $H_c = \{h_j^c\}_{j \in \mathcal{V}_c}$. We add learned type-specific offsets $s_r, s_c \in \mathbb{R}^d$ to distinguish rows/cols and concatenate:

$$\mathcal{T} = \big\{h_i^r + s_r\big\}_i \cup \big\{h_j^c + s_c\big\}_j \in \mathbb{R}^{N \times d}, \quad (12)$$

with $N = |\mathcal{V}_r| + |\mathcal{V}_c|$ and model width $d = \texttt{emb\_size}$. During training, we apply token dropout with rate $p$ while preserving expected magnitude and at least one token per graph.

### B.2.1    PERCEIVER-LIKE ATTENTION BLOCK

We introduce three learnable latent queries (*type*, *scale*, *complexity*), $Q^{(0)} = \big[q_{\text{type}}, q_{\text{scale}}, q_{\text{comp}}\big] \in \mathbb{R}^{3 \times d}$. For extremely large graphs we perform a light-weight Top-$K$ truncation of context tokens by their $\ell_2$-norm:

$$\mathcal{T}_K = \arg \text{topK} \|t\|_2, \qquad K = \texttt{max\_token\_attn}, \quad (13)$$
$$\quad\quad{}_{t \in \mathcal{T}}$$

and use $\mathcal{T}_K$ as keys/values.

Each Perceiver block consists of cross-attention from latents to tokens, followed by latent self-attention and an MLP:

$$\tilde{Q}^{(\ell)} = Q^{(\ell-1)} + \text{DropPath}\Big(\text{Drop}\Big(\text{Attn}\Big(\text{LN}(Q^{(\ell-1)}), \text{LN}(\mathcal{T}_K), \text{LN}(\mathcal{T}_K)\Big)\Big)\Big), \quad (14)$$

$$\hat{Q}^{(\ell)} = \tilde{Q}^{(\ell)} + \text{DropPath}\Big(\text{Drop}\Big(\text{Attn}\Big(\text{LN}(\tilde{Q}^{(\ell)}), \text{LN}(\tilde{Q}^{(\ell)}), \text{LN}(\tilde{Q}^{(\ell)})\Big)\Big)\Big), \quad (15)$$

$$Q^{(\ell)} = \hat{Q}^{(\ell)} + \text{DropPath}\Big(\text{MLP}\Big(\hat{Q}^{(\ell)}\Big)\Big), \quad (16)$$

where multi-head attention is

$$\text{Attn}(Q, K, V) = \text{Concat}\big(\text{head}_1, \ldots, \text{head}_H\big) W^O, \text{head}_h = \text{Softmax}\left(\frac{QW_h^Q (KW_h^K)^\top}{\sqrt{d/H}}\right) VW_h^V.$$
$$(17)$$

The block outputs three refined latent embeddings $z_{\text{type}}, z_{\text{scale}}, z_{\text{comp}} \in \mathbb{R}^d$ from $Q^{(L)}$.

**Why Perceiver-like block instead of Full Self-Attention?**    A naive self-attention over all node tokens scales quadratically in $N$: time/memory $\Theta(N^2 d)$, which is prohibitive for MILPs where $N$ can reach tens of thousands. In contrast, the Perceiver-style cross-attention uses only $L_q = 3$ latent queries and $K \ll N$ context tokens:

$$\text{cost} = \underbrace{\Theta(L\, L_q K d)}_{\text{cross-attn}} + \underbrace{\Theta(L\, L_q^2 d)}_{\text{latent self-attn}} \quad \ll \quad \Theta(N^2 d),$$

and remains linear in $K$ while preserving targeted aggregation via learned queries. Top-$K$ truncation further caps compute/memory without architectural changes, and token dropout regularizes training on large instances.

### B.2.2    PROJECTION HEADS AND OUTPUTS

From the disentangled latents we compute task-specific outputs with lightweight MLP heads:

$$y_{\text{type}} = f_{\text{type}}(z_{\text{type}}) = W_2 \,\text{Drop}(\text{GELU}(W_1 \,\text{LN}(z_{\text{type}}))) \in \mathbb{R}^{4096}, \quad (18)$$

$$\hat{s} = f_{\text{scale}}(z_{\text{scale}}) \in \mathbb{R}^8, \qquad \hat{t} = f_{\text{comp}}(z_{\text{comp}}) \in \mathbb{R}, \quad (19)$$

where the **type head** projects to a 4096-d space used for contrastive alignment with a precomputed 4096-d text embedding; the **scale head** regresses eight log-statistics; and the **complexity head** regresses log-solving time. Following CLIP, we maintain a learnable temperature $\tau > 0$ (implemented as $\log \tau$) to scale similarities:

$$\mathbf{S}_{m \to t} = \tau \cos\left(\frac{y_{\text{type}}}{\|y_{\text{type}}\|_2}, \frac{y_{\text{text}}}{\|y_{\text{text}}\|_2}\right), \qquad \mathbf{S}_{t \to m} = \tau \cos\left(\frac{y_{\text{text}}}{\|y_{\text{text}}\|_2}, \frac{y_{\text{type}}}{\|y_{\text{type}}\|_2}\right). \qquad (20)$$

These heads correspond directly to the three losses detailed in Appendix B.2.3.

### B.2.3 Training Objectives

To enforce the desired disentanglement, we train GLIM using a composite loss function that combines objectives for each of the three factors. The total loss is a weighted sum of the individual losses:

$$\mathcal{L} = \lambda_{\text{type}}\mathcal{L}_{\text{type}} + \lambda_{\text{scale}}\mathcal{L}_{\text{scale}} + \lambda_{\text{comp}}\mathcal{L}_{\text{comp}}, \qquad (21)$$

where $\lambda_{\text{type}}$, $\lambda_{\text{scale}}$ and $\lambda_{\text{comp}}$ are hyperparameters that balance the contribution of each task.

**Type Objective**  Following the approach of CLIP (Radford et al., 2021), for each MILP instance in a batch, we have its corresponding type label, which is a text embedding derived from the problem's formulation code (see Appendix A.1.2). The goal is to maximize the cosine similarity between the MILP's type embedding and its paired text embedding, while simultaneously minimizing the similarity with all other text embeddings in the same batch. This is achieved through a symmetric cross-entropy loss over the similarity scores.

For a batch of $N$ pairs, the loss is:

$$\mathcal{L}_{\text{type}} = \frac{1}{2}\left(\text{CE}(\mathbf{S}_{m \to t}) + \text{CE}(\mathbf{S}_{t \to m})\right),$$

where $\mathbf{S}_{m \to t}$ and $\mathbf{S}_{t \to m}$ are the $N \times N$ matrices of cosine similarities between all MILP and text embeddings in the batch, scaled by a learnable temperature parameter. CE denotes the cross-entropy loss with respect to identity-matrix labels.

**Scale Objective**  The scale objective is a multi-target regression task. The scale head, an MLP, takes the scale embedding $z_{\text{scale}}$ as input and predicts a vector of 8 structural statistics (e.g., number of variables, constraints). As these statistics often follow a heavy-tailed distribution, we predict their logarithmic values. The loss is the Mean Squared Error (MSE) between the predicted and ground-truth log-statistics:

$$\mathcal{L}_{\text{scale}} = \frac{1}{8}\sum_{i=1}^{8}\left(\text{MLP}_{\text{scale}}(z_{\text{scale}})_i - \log(\hat{s}_i + \epsilon)\right)^2, \qquad (22)$$

where $s_i$ is the $i$-th ground-truth statistic and $\epsilon$ is a small constant for numerical stability.

**Complexity Objective**  Similarly, the complexity objective is a regression task. The complexity head predicts the solver time for the instance based on the complexity embedding $z_{\text{comp}}$. As with the scale labels, we use the log-transformed solving time as the target to mitigate the effect of outliers and stabilize training. The loss is also defined as the Mean Squared Error:

$$\mathcal{L}_{\text{comp}} = \left(\text{MLP}_{\text{comp}}(z_{\text{comp}}) - \log(\hat{t} + \epsilon)\right)^2, \qquad (23)$$

where $\hat{t}$ is the ground-truth solving time. This multi-task objective forces the model to encode distinct, interpretable information into each component of the final embedding vector.

### B.3 Implementation Details

We use `DistributedDataParallel` for multi-GPU training. The largest model, GLIM-L, was trained on 4 NVIDIA V100 GPUs for approximately three days. The hyperparameters used in our experiments are reported in Table 8.

Table 8: Training hyperparameters for GLIM models.

| Hyperparameter | Value |
|---|---|
| Batch size | 8 |
| Epochs | 100 |
| Learning rate | 0.00005 |
| Dropout | 0.2 |
| Attention dropout | 0.1 |
| Drop path | 0.05 |
| Token dropout | 0.1 |
| Type loss weight | 2.0 |
| Scale loss weight | 2.0 |
| Complexity loss weight | 1.0 |
| Embedding size | 256 |
| Number of attention layers | 4 |
| Number of GNN layers | 2 |
| GAT heads | 16 |
| Attention heads | 16 |
| Max token attention | 16384 |

## C  DETAILS OF MILP SOLVER HYPERPARAMETER PREDICTION

To train the MLP head for solver hyperparameter prediction, we first needed to construct a suitable training dataset. The input data of the training set is identical to that used by the GLIM backbone; the key difference lies in the labels, which correspond to the near-optimal solver hyperparameters for each instance. To obtain these labels, we employed `smac3` (Lindauer et al., 2022), a widely used Bayesian optimization framework, to perform hyperparameter tuning on the Gurobi solver. Gurobi provides a wide range of hyperparameter interfaces. From these, we use the same hyperparameter selection space as in (Guo et al., 2024), as summarized in Table 9, which includes parameters governing MIP strategy, simplex procedures, presolving, cut generation, and other algorithmic components of Gurobi.

Since hyperparameter tuning with `smac3` is computationally expensive, we restricted the tuning process to instances with an original solving time (i.e., under Gurobi default settings) of less than 5 seconds. For each eligible instance, tuning was capped at 200 trials, and the best hyperparameter configuration found was recorded. We further filtered out instances for which no speedup was achieved, thereby obtaining the final dataset used for training the MLP head.

Table 9: Selected hyperparameters of Gurobi. The column "category" indicates the solver component affected by the hyperparameter.

| Hyperparameter | Category | Value Type | Range | Description |
|---|---|---|---|---|
| Heuristics | MIP | double | [0, 1] | Controls the intensity of MIP heuristics. |
| MIPFocus | MIP | integer | {0, 1, 2, 3} | Sets the focus of the MIP solver. |
| VarBranch | MIP | integer | {-1, 0, 1, 2, 3} | Variable branching strategy. |
| BranchDir | MIP | integer | {-1, 0, 1} | Preferred branching direction. |
| RINS | MIP | integer | {-1, 0, …, 20} | RINS heuristic level. |
| PartitionPlace | MIP | integer | {0, 1, …, 31} | Controls when the partition heuristic is applied. |
| NodeMethod | MIP | integer | {-1, 0, 1, 2} | Method for solving MIP node relaxations. |
| LPWarmStart | Simplex | integer | {0, 1, 2} | Warm start usage in simplex. |
| PerturbValue | Simplex | double | [0, 0.001] | Magnitude of simplex perturbation. |
| Presolve | Presolve | integer | {-1, 0, 1, 2} | Presolve aggressiveness level. |
| Prepasses | Presolve | integer | {-1, 0, …, 20} | Maximum number of presolve passes. |
| Cuts | MIP Cuts | integer | {-1, 0, 1, 2, 3} | Global cut generation control. |
| CliqueCuts | MIP Cuts | integer | {-1, 0, 1, 2} | Clique cut generation level. |
| CoverCuts | MIP Cuts | integer | {-1, 0, 1, 2} | Cover cut generation level. |
| Method | Other | integer | {-1, 0, 1, 2, 3, 4, 5} | Algorithm for solving continuous models. |

The loss function of the MLP head is defined as the prediction error over the selected solver hyperparameters. For integer-type hyperparameters, we employ cross-entropy loss, while for double-type

hyperparameters we use mean squared error (MSE). The overall training objective is obtained by summing the losses of all hyperparameters with equal weights.

The input to the MLP head is a 768-dimensional vector, constructed by concatenating three 256-dimensional embedding tokens that encode *type*, *scale*, and *complexity*, respectively. During inference, predictions for double-type hyperparameters are taken directly as continuous outputs, whereas integer-type hyperparameters are decoded using greedy selection. The hyperparameters used for training the MLP head are summarized in Table 10.

Table 10: Hyperparameters for training the MLP head.

| Hyperparameter | Value |
|---|---|
| Hidden Dimension | 256 |
| Learning Rate | $1.0 \times 10^{-3}$ |
| Label Smoothing | 0.1 |
| Epochs | 100 |
| Batch Size | 8 |

## D  FURTHER EXPERIMENTS AND ANALYSIS

### D.1  ABLATION STUDY

We compare the Perceiver-like attention block used in GLIM with a standard self-attention block. In prior work Li et al. (2025), self-attention blocks were directly applied to the node embeddings produced by a GNN in the context of Language–MILP Contrastive Learning. To enable a fair comparison, we replace GLIM's Perceiver-like block with a self-attention block. Since self-attention requires $O(n^2)$ attention computation, we cap the maximum token input length at 512; for larger MILP instances, we subsample the node embeddings. We evaluate the impact of this architectural choice on the MILP Instance Retrieval task, with results reported in Table 11.

Table 11: Comparison of MEE results on MILP Instnace Retrieval task of different GLIM model architectures.

|  | Target=Scale | | Target=Complexity | |
|---|---|---|---|---|
| Dataset | GLIM-valid | MIPLIB | GLIM-valid | MIPLIB |
| GLIM-S w/ Self-Attention | 0.121 | 1.547 | 0.869 | 3.721 |
| GLIM-S | 0.038 | 1.442 | 0.671 | 4.053 |
| GLIM-M w/ Self-Attention | 0.054 | 1.306 | 0.726 | 3.760 |
| GLIM-M | 0.026 | 1.255 | 0.678 | 3.494 |
| GLIM-L w/ Self-Attention | 0.030 | 0.892 | 0.680 | 3.746 |
| GLIM-L | 0.024 | 1.203 | 0.645 | 3.357 |

### D.2  MORE RESULTS ON MILP SOLVER HYPERPARAMETER PREDICTION

We evaluate the performance of GLIM variants and baseline models on four in-distribution benchmark problems: Combinatorial Auction (CA), Generalized Independent Set (GISP), Fixed Charge Multi-Commodity Network Flow (FCMCNF), and Set Cover (SC). Results for Win Rate are reported in Table 12, while comparisons on Average Solving Time and Speedup Ratio are presented in Table 13. As shown, GLIM-S outperforms both GLIM-M and GLIM-L. This is because the training corpus of GLIM-S already includes these benchmark problems, whereas GLIM-M and GLIM-L are trained on much larger and more diverse datasets, which dilutes their representation power for problems already seen in the training distribution.

Table 12: Win rate results for the *MILP Solver Hyperparameter Prediction* task.

| Model | In-Distribution | | | |
|---|---|---|---|---|
| | CA | GISP | FCMCNF | SC |
| Linq-Embed-Mistral | 50/50 | 44/44 | 31/37 | 50/50 |
| Qwen-3-Embed | 50/50 | 44/44 | 34/37 | 50/50 |
| GLIM-S | 50/50 | 44/44 | 33/37 | 50/50 |
| GLIM-M | 50/50 | 44/44 | 33/37 | 50/50 |
| GLIM-L | 50/50 | 44/44 | 32/37 | 50/50 |

Table 13: Average solving time results for the *MILP Solver Hyperparameter Prediction* task. The ratio in parentheses is the average speedup ratio.

| Model | In-Distribution | | | |
|---|---|---|---|---|
| | CA | GISP | FCMCNF | SC |
| Default | 69.715 | 28.068 | 29.727 | 15.380 |
| Linq-Embed-Mistral | 15.357 (9.77) | 3.207 (12.36) | 8.829 (3.11) | 3.055 (8.96) |
| Qwen-3-Embed | 18.513 (7.92) | 2.385 (14.22) | 8.206 (3.60) | 2.587 (9.46) |
| GLIM-S | 3.959 (21.09) | 1.809 (17.28) | 7.803 (3.77) | 2.871 (9.16) |
| GLIM-M | 6.502 (20.56) | 4.086 (13.19) | 8.150 (3.31) | 3.013 (8.38) |
| GLIM-L | 5.600 (11.60) | 2.461 (12.37) | 8.896 (2.68) | 4.284 (6.27) |

