# OpenReview forum: "GLIM: Towards Generalizable Learning Representation for MILP"
_ICLR.cc/2026/Conference — Submitted to ICLR 2026_

### Official Review · Reviewer_LSsY · 2025-10-29

**Soundness:** 3
**Presentation:** 4
**Contribution:** 3
**Rating:** 6
**Confidence:** 4

**Summary:**

This paper proposes a general-purpose embedding model that learns a joint representation to disentangle type, scale, and solving complexity of mixed integer linear programs (MILPs). The proposed architecture consists of (1) a bipartite GNN encoder to embed the MILP instance (2) a perceiver like mix-attention block that performs cross attention between three learnable latent vectors (type, scale, and complexity token) with all node embeddings, and (3) final projection head. The authors use type loss, scale loss, and complexity loss to encode complementary and interpretable factors within the embedding. The authors finally demonstrate the effectiveness of the architecture on MILP instance retrieval and MILP solver hyper-parameter prediction.

**Strengths:**

1. This is a well-written paper.

2. The idea of disentangling type, scale and complexity of MILP instances make a lot of sense, and to my knowledge, is quite novel.

3. Empirical results on MILP instance retrieval and MILP solver hyperparameter prediction seems convincing.

**Weaknesses:**

1. The baselines compared in the paper (Linq-Embed-Mistral and Qwen-3-Embed) are purely LLM-based text embedding model and seems to be quite weak. There has been many linear attention architectures proposed in previous literature (e.g. the simplest would be self-attention with sliding windows). To strengthen the results, the author should consider comparing with those linear attention variants.

2. Empirical experiments are only performed on graph-level prediction tasks. To strengthen the results, the author should consider node-level or edge-level prediction task (e.g. learning to branch, learning to cut) to show that the disentanglement loss with the proposed architecture can help generally for many learning for MILP tasks.

**Questions:**

See the weaknesses above. I have the following question to expand upon Weakness 1:

1. Appendix D.1 architecture ablation. The authors cap the maximum token for self-attention to 512, which seem restrictive. How would self-attention perform with a larger maximum token limit? Furthermore, how would the self-attention baseline perform for hyper-parameter prediction?

---

### Official Review · Reviewer_nMh7 · 2025-10-29

**Soundness:** 2
**Presentation:** 2
**Contribution:** 3
**Rating:** 2
**Confidence:** 4

**Summary:**

This paper introduces GLIM (Generalizable Learning Representation for MILPs), a general-purpose embedding model for Mixed-Integer Linear Programs. The authors argue that existing learning-based MILP solvers lack generalization across tasks and distributions, and propose to learn a disentangled embedding that captures three orthogonal factors — problem type, scale, and complexity. Each MILP instance is represented as a bipartite graph and encoded through a hybrid GNN–Perceiver architecture. The model is trained on a large synthetic dataset with multi-task supervision. Two downstream tasks — MILP instance retrieval and solver hyperparameter prediction — are used to evaluate the learned representation, showing some improvement over text-based embedding baselines.

**Strengths:**

The paper tackles an important and under-explored direction: learning general-purpose representations for MILP problems, rather than training specialized models for a single solver component or problem class. This line of work is timely and could serve as a foundation for future multi-task or cross-domain learning in combinatorial optimization.

**Weaknesses:**

1. The paper lacks meaningful comparison with existing learning-based solvers that also involve instance embeddings. For example, VAE-based models like G2MILP [1], multimodal approaches such as MILP-Evolve [2], and learning-based solvers like Predict & Search [3] all involve representation learning of MILPs, though they are not focused on representation learning directly. A recent work [4] also explores cross-domain MILP training. The authors should position their contribution more clearly with respect to these works and provide quantitative or qualitative comparisons when possible.
1. The proposed decomposition of MILP representation into type, scale, and solving complexity appears intuitively reasonable but lacks empirical justification. These three factors are not truly orthogonal, as solving complexity is influenced by both type and scale, and the paper provides no analysis demonstrating independence or disentanglement among them. Moreover, the learned representation is global (one vector per instance) and thus may not generalize to node-level or constraint-level tasks (e.g., branching, diving, cut selection, or solution prediction), which limits practical applicability.
1. The evaluation tasks are quite limited and largely self-defined. Instance retrieval and hyperparameter prediction are not standard benchmarks in the ML4CO literature, and thus do not convincingly demonstrate the utility of the learned embeddings. There are no strong baselines available for these tasks, making it difficult to assess the real advantage of GLIM over simpler alternatives. Applying the learned representations to more realistic solver-level tasks (such as heuristic selection, branching, or solution prediction) would significantly strengthen the work.
1. From Tables 3 and 4, while GLIM achieves improvement over text-based embeddings, the gains are generally small and sometimes inconsistent across datasets. Given that the baselines are general text embedding models rather than architectures designed for MILPs, the empirical advantage appears marginal and does not yet support the claim of a robust, generalizable representation.

[1] Geng et al., A deep instance generative framework for milp solvers under limited data availability. NeurIPS 2023.

[2] Li et al., Towards foundation models for mixed integer linear programming. ICLR 2025.

[3] Han, et al. A GNN-Guided Predict-and-Search Framework for Mixed-Integer Linear Programming. ICLR. 2023.

[4] https://openreview.net/pdf?id=wRQmQ6UXYF

**Questions:**

See Weaknesses.

---

### Official Review · Reviewer_v3fD · 2025-10-30

**Soundness:** 2
**Presentation:** 2
**Contribution:** 2
**Rating:** 2
**Confidence:** 4

**Summary:**

This work tackles the compelling problem of learning a generalizable representation for MILPs. While the direction is timely and important, the empirical evaluation in its current state does not adequately substantiate the core claims. The performance improvements over the baseline are marginal. Furthermore, the claim of generalizability is critically underexplored, resting on a single meaningful downstream task. To convincingly demonstrate a general-purpose representation, the work must show its effectiveness across a broader spectrum of fundamental MILP tasks, such as branching, cutting plane selection, and—most importantly—generalization to unseen problem types.

**Strengths:**

The graph representation is interesting.

**Weaknesses:**

I have the following concerns:

- While the authors claim their framework is highly generalizable, the choice of downstream tasks does not sufficiently support this. The MILP Instance Retrieval task is non-standard and its relevance to practical solver performance is unclear. To convincingly demonstrate generalizability, the framework should be evaluated on core, well-established tasks such as learning to branch, solution prediction, or learning to cut.
- Furthermore, the evaluation of the hyper-parameter prediction task raises concerns. On MIPLIB, GLIM outperforms the default solver in only half the instances, suggesting limited practical advantage. More critically, the use of the "Moderated Solving Time" metric is misleading, as it artificially inflates performance by taking the minimum of two runtimes. For a fair assessment, the raw runtime using the predicted hyper-parameters must be reported.

**Questions:**

- In the anonymous link, I do not see the code for the task of instance retrieval.

---

### Official Review · Reviewer_x1Dh · 2025-10-30

**Soundness:** 2
**Presentation:** 3
**Contribution:** 3
**Rating:** 4
**Confidence:** 2

**Summary:**

This paper introduces an embedding model guided by three factors: problem type, scale, and complexity for mixed-integer linear programs (MILPs). The primary objective is to leverage the learned embeddings to enhance the generalization performance of downstream tasks. The authors provide empirical evidence of the advantages of their approach over baseline methods, although the baselines and considered instances could be strengthened to better demonstrate the method’s capabilities.

**Strengths:**

1. The paper is well-written and presents a clear, logical flow of ideas.
2. The concept of using embedding learning to enhance generalization performance is both reasonable and intriguing.
3. The empirical results show improvements compared to the selected baselines.

**Weaknesses:**

Below are my major concerns:

1. **Simplistic instances**: While the MILP instances used in the embedding experiments are large (up to 100k variables and 100k constraints), the filtering process retains only those instances that can be solved optimally within 1000 seconds. These instances are relatively simple compared to real-world MILPs, where even solving for hours or days may not guarantee an optimal solution. Similarly, the datasets used for hyperparameter prediction are relatively easy and can be solved by the default solver in an average of less than 100s (some even in under 5s). I recommend incorporating more challenging datasets to demonstrate the robustness of the approach in real-world scenarios.
2. **Limited baselines**: The hyperparameter tuning task is evaluated only against other embedding approaches. I would expect comparisons that demonstrate the effectiveness of the proposed embeddings when applied to existing hyperparameter tuning methods, which would offer a stronger evaluation.

Since I am not deeply familiar with the field of generalization and embedding learning, I express a relatively low level of confidence in my review.

**Questions:**

Please see the weaknesses.

---

### Meta-Review · Area_Chair_12uK · 2025-12-31

**Summary:**

While the reviewers acknowledged the paper's clear writing and the interesting motivation behind disentangling problem type, scale, and complexity for MILP representation, there is a consensus that the current empirical evaluation and methodological positioning are insufficient to support the strong claims of "generalizability." The decision is to recommend rejection (marginally below threshold).

Although the proposal of a unified embedding model for MILPs is timely, the current execution does not convincingly demonstrate distinct advantages over existing methods due to the selection of weak baselines and non-standard evaluation tasks. The reviewers encourage the authors to extend the framework to widely accepted solver tasks and compare against domain-specific state-of-the-art baselines to substantiate the generalization claims.

**Reviewer Concerns:**

The author gave up discussing with reviewers during the rebuttal process. The decision to recommend rejection (marginally below threshold) is primarily informed by the following concerns:

1. Multiple reviewers (v3fD, nMh7, LSsY) expressed significant concern that the chosen downstream tasks—Instance Retrieval and Hyperparameter Prediction—are not standard benchmarks in the "Machine Learning for Combinatorial Optimization" (ML4CO) literature. Reviewers noted that to truly demonstrate a "general-purpose" representation, the model should be evaluated on core solver tasks such as learning to branch, cut selection, or solution prediction (node/edge-level tasks). The relevance of "Instance Retrieval" to practical solver performance remains unclear to the reviewers.

2. The reviewers found the comparative analysis lacking in rigor.

- Strawman Baselines: Comparing GLIM primarily against LLM-based text embeddings (e.g., Mistral, Qwen) was seen as a weak baseline (LSsY, nMh7), given that these are not designed for structural MILP data.

- Missing SOTA: Reviewer nMh7 pointed out a lack of comparison with existing learning-based MILP frameworks (e.g., G2MILP, MILP-Evolve, or Predict & Search) that also utilize instance representations.

- Task-Specific Baselines: For the hyperparameter prediction task, Reviewer x1Dh noted the absence of comparisons to established hyperparameter tuning methods, limiting the assessment of practical utility.

3. Concerns regarding Dataset Difficulty and Metrics: The validity of the empirical results was questioned on two fronts:

- Instance Difficulty: Reviewer x1Dh highlighted that the dataset filtering process (retaining instances solvable <1000s, with some HPO datasets solvable <5s) results in "simplistic" instances that do not reflect the complexity of real-world MILPs.

- Misleading Metrics: Reviewer v3fD criticized the "Moderated Solving Time" metric used in hyperparameter prediction, arguing it artificially inflates performance. They suggested that raw runtime improvements are necessary for a fair assessment.

4. Reviewer nMh7 raised theoretical concerns regarding the central hypothesis of the paper. The assumption that type, scale, and complexity are "orthogonal" factors lacks empirical justification or independence analysis. Furthermore, the global nature of the learned representation (one vector per instance) is seen as a bottleneck that inherently limits the model's applicability to the node-level tasks (branching/cutting) required for a truly generalizable MILP backbone.

**Reviewer Scores:**

No reviewer would have changed their score since the authors gave up discussing during the rebuttal process.

---

### Decision · Program_Chairs · 2026-01-26

Reject